# Comparative Impact of PD-1 and PD-L1 Inhibitors on Advanced Esophageal or Gastric/Gastroesophageal Junction Cancer Treatment: A Systematic Review and Meta-Analysis

**DOI:** 10.3390/jcm10163612

**Published:** 2021-08-16

**Authors:** SuA Oh, Eunyoung Kim, Heeyoung Lee

**Affiliations:** 1Evidence-Based and Clinical Research Laboratory, Department of Health, Social and Clinical Pharmacy, College of Pharmacy, Chung-Ang University, Seoul 06974, Korea; dhtndk0@cau.ac.kr; 2Department of Clinical Medicinal Sciences, Konyang University, Nonsan 32992, Korea

**Keywords:** PD-1/PD-L1 inhibitor, gastric esophageal cancer, meta-analysis

## Abstract

Programmed death 1 (PD-1) and PD ligand 1 (PD-L1) inhibitors have demonstrated varying effectiveness in treating esophageal or gastric/gastroesophageal junction (G/GEJ) cancer. Hence, this systematic review and meta-analysis evaluated the efficacy and safety of anti-PD-1/PD-L1 treatment in patients with esophageal or G/GEJ cancer by analyzing the types of medications. Randomized controlled trials comparing anti-PD-1/PD-L1 to control therapy were identified by searching PubMed, EMBASE, and ClinicalTrials.gov. The outcomes included overall survival (OS), progression-free survival (PFS) rates, and serious adverse events (SAEs), evaluating the differences in therapy types, including a comparison between PD-1 and PD-L1 inhibitors. Eight studies were included in the analysis. PD-1/PD-L1 inhibitors affected the overall OS rate increment without influencing the PFS rate (HR, 0.837; 95% CI, 0.753–0.929; *p* = 0.001; HR 0.991; 95% CI, 0.778–1.263; *p* = 0.942, respectively). Anti-PD-1 was significantly more beneficial for increasing OS and PFS than PD-L1 inhibitors. Anti-PD-1 and PD-L1 use was not significantly associated with SAE development in esophageal or G/GEJ cancer patients. PD-1/PD-L1 inhibitor use was associated with improved OS and PFS rate increase among PD-1 and PD-L1 inhibitors. Considering response variations to anti-PD-1/PD-L1 usage, more individualized treatments should be introduced in clinical practice.

## 1. Introduction

Esophageal or gastric/gastroesophageal junction (G/GEJ) cancer is one of the most fatal cancers [1,2]. Although these cancer types could not be together considered, controversies remain to differentiate esophageal or G/GEJ in terms of histological distinctions and clinical outcomes [3]. In addition, recently, a study was also conducted that examined esophageal and gastroesophageal junction cancer together [4]. Typically, esophageal or G/GEJ cancer is often asymptomatic in the early stages, resulting in a loss of opportunity for appropriate treatments in advanced cases. Still, recommended therapies primarily include surgical or systemic treatment such as platinum-based chemotherapy, the five-year survival rate remains around 15–25%, and high recurrence or metastasis rates have been reported [5]. Pathophysiologically, gastrointestinal tumors are traditionally considered non-immune-related malignancies. However, current reports have provided a new direction in the treatment by demonstrating the effectiveness of blocking specific immunosuppressive substances, such as programmed death receiver 1/programmed death ligand 1 (PD-1/PD-L1) [6]. Current studies have indicated that the number of tumor-infiltrating lymphocytes is related to tumor prognosis [7] and the correlation between malignant tumors and immune cells such as T cells. Thus, in order to enhance immune activity against cancer cells, PD-1/PD-L1 inhibitors have been developed. With regard to pathophysiological ensurance, various clinical studies have been conducted to provide empirical evidence for evaluating the effects of PD-1/PD-L1 inhibitors on esophageal or G/GEJ cancer—a cancer type with a low survival rate [6,8]. However, even the outcomes of systematic reviews and meta-analysis studies have not been able to provide conclusive information on the clinical benefits of PD-1/PD-L1 inhibitor usage for treatment of esophageal or G/GEJ cancer, as opposed to chemotherapies [6,8,9,10]. A meta-analysis conducted by Wang et al. [8] suggested that treating esophageal or G/GEJ cancer with PD-1/PD-L1 inhibitors resulted in a higher pooled hazard ratio (HR) for overall survival (OS) and progression-free survival (PFS), indicating their ineffectiveness. In contrast, Chen et al. [6] reported that PD-1/PD-L1 inhibitors were more effective than control therapy in the treatment of esophageal or G/GEJ cancer, demonstrating increased pooled odds ratios (ORs) for OS and PFS. Nonetheless, considering the limited number of studies conducted and inconsistencies in study designs in meta-analyses resulting in significantly heterogeneous results [6,8,9,10,11], more well-designed systematic studies—including randomized controlled trials (RCTs)—are necessary to further verify the efficacy and safety of PD-1/PD-L1 inhibitors in treating esophageal or G/GEJ cancer. Furthermore, as one of the standard methods in use for analyzing survival outcomes in oncology trials, HR in OS and PFS should be measured as endpoints in more statistically powerful studies [12]. Although currently a study demonstrated the efficacy of HR in OS and PFS in the treatment of PD-1/PD-L1 based on the regimen, only a network meta-analysis was available [13]. Since multiple indirect comparison trials are required to produce a sufficient level of power and precision as a single head-to-head trial [14], performing a future meta-analysis with head-to-head trials on the efficacy of anti-PD-1/PD-L1 for esophageal or G/GEJ cancer treatment is essential.

Moreover, in order to select and develop optimal treatments, a comprehensive understanding of the potential discrepancies between PD-1 and PD-L1 inhibitors could prove clinically significant [15]. Taking into consideration the different mechanisms of action of anti-PD-1 and anti-PD-L1, the clinical discrepancies between anti-PD-1 and anti-PD-L1 for treating patients with esophageal or G/GEJ cancer should be sufficiently evaluated using statistically powerful methods [16]. In particular, although all PD-1 and PD-L1 inhibitors are commonly reported to function in the same signaling pathway, the clinical efficacy of single agents remains unclear [15]. Thus, an overall comparison of the efficacy and safety between PD-1/PD-L1 inhibitors and conventional therapies to treat esophageal or G/GEJ cancer, along with indirect evaluation of the differences among PD-1/PD-L1 inhibitors, was conducted in this study via a systematic review and meta-analysis.

## 2. Materials and Methods

This meta-analysis was conducted in accordance with the Preferred Reporting Items for Systematic Reviews and Meta-Analyses (PRISMA) guidelines [17] (Appendix A), and the protocol was registered on the International Prospective Register of Systematic Reviews database under no. CRD42021231799.

### 2.1. Data Sources and Search Strategy

The PubMed and EMBASE databases were searched up to December 2020 to select eligible articles using a comprehensive search strategy with relevant keywords. ClinicalTrials.gov (accessed on 30 December 2020) a registry and results database of clinical trials, was also searched to assess potential publication bias and identify ongoing trials. A manual search was conducted to detect clinical trials of various PD-1/PD-L1 inhibitors. Titles and abstracts were distinguished using the following search terms to sort out relevant text: “Randomized controlled trial”, “Gastric or gastroesophageal junction cancer”, “Esophageal Cancer”, “Pembrolizumab,” Nivolumab”, “Camrelizumab”, and “Avelumab”. The search strategy has been described further in Appendix A.

### 2.2. Study Selection

Two independent investigators first evaluated the titles and abstracts of the retrieved literature to identify potentially relevant articles. For inclusion in this study, the selected literature was required to meet all of the following criteria: (1) all studies must consist of RCTs; (2) all subjects must have advanced esophageal or G/GEJ cancer; (3) the intervention group was treated with PD-1/PD-L1 inhibitor without combination except standard of care (SOC) treatment; (4) the control group was selected whose definition was an SOC treatment recommended by the National Comprehensive Cancer Network (NCCN) guidelines [18] or a placebo. Each study should also have provided data on efficacy or safety. Case reports, single-arm trials, observational studies, and animal studies were not included. Additionally, studies that included a sample size of less than five people, articles written in languages other than English, and studies in which all contents could not be verified were excluded from the analysis. If there was a disagreement between the two investigators, the agreement was mutually resolved.

### 2.3. Data Extraction and Quality Assessment

Two independent reviewers extracted the data from the selected literature. Initially, information about the first author, year of publication, number of participants, interventions, clinical trial phase, study design, treatment line, tumor site, the primary endpoint of the included study, and patient age was obtained. Efficacy data—such as OS and PFS—were subsequently collected. The primary efficacy outcome was OS, secondary efficacy outcome was PFS. Finally, safety data such as serious adverse events (SAEs) were assessed. The quality of the RCTs was evaluated using a bias risk assessment tool developed by the Cochrane Collaboration [19]. The rationale for the RCT study was estimated as a Grading of Recommendations, Assessment, Development, and Evaluations profiler (GRADEpro) approach—classified as high, medium, low, or very low [20]. Any disagreements between the two investigators were resolved through discussion.

### 2.4. Statistical Analyses (Data Synthesis and Analysis)

To measure the efficacy of PD-1/PD-L1 inhibitors, OS and PFS data were evaluated using HRs and 95% confidence intervals (CIs). For the subgroup analysis, the included studies were arranged; anti-PD-1 and PD-L1, individual agents, treatment line, and tumor site were estimated. Using the previously developed engage-digitizer and calculator, HR from Kaplan–Meier curves was obtained if there was no prior indication in the included studies [21]. Additionally, the safety outcomes were assessed by measuring the number of developed SAEs.

The I^2^ statistic was used to evaluate heterogeneity among studies, and the percentile statistics were classified as low (<25%), medium (25–50%), or high (>50%) [22]. Furthermore, publication bias was estimated by examining the funnel plots and Egger’s weighted regression test [23]. Statistical significance was set at *p* < 0.05. Regardless of the observed statistical heterogeneity, pre-specified subgroup analyses were performed. All statistical analyses were performed using RevMan (version 5.4; The Nordic Cochrane Center, The Cochrane Collaboration, Copenhagen, Denmark, 2020) and Comprehensive Meta-Analysis (CMA) (version 2; Biostat, Englewood, NJ, USA).

## 3. Results

### 3.1. Study Selection

The comprehensive search identified 1015 potentially eligible studies from the databases and registry reports assessed for eligibility, wherein 124 articles were selected. After excluding 116 articles, eight were included in our final analysis (Figure 1). A manual search of the reference lists yielded no additional relevant studies.

### 3.2. Study Characteristics

Table 1 presents the basic characteristics of the eight studies selected for inclusion [24,25,26,27,28,29,30,31]. In total, 4206 patients were included. Pembrolizumab (PEM) was administered at 200 mg every 3 weeks to 1239 patients across four studies [25,28,29,31]. Nivolumab (NIV) was administered to 540 patients at 240 mg every 2 weeks and 3 mg/kg every 2 weeks, respectively, in two studies [24,27]. Similarly, 185 patients in one study were treated with 10 mg/kg avelumab (AVE) every 2 weeks [26]. In yet another study, a total of 228 patients received 200 mg of camrelizumab (CAM) every 2 weeks [30].

### 3.3. Efficacy Outcomes

#### 3.3.1. OS

For primary efficacy outcome, eight studies in total were included in the current analysis for the assessment [24,25,26,27,28,29,30,31]. In OS, the group treated with the anti-PD-1/PD-L1s, a lower HR was identified than in the control group (HR 0.837, 95% CI: 0.753–0.929, I^2^ = 63.132%, *p* = 0.001, Figure 2a). Subgroup analysis comparing between the anti-PD-1 and anti-PD-L1 groups showed significant differences (*p* = 0.016), demonstrating that a lower HR was reported in the anti-PD-1 group as opposed to the anti-PD-L1 group (HR 0.800, 95% CI: 0.726–0.882, *p* < 0.001 vs. HR 1.114, 95% CI: 0.867–1.431, *p* = 0.397, respectively, Figure 2b). Upon comparing each agent as shown in Figure 2c, the HR exhibited variation in the group of patients receiving AVE (HR 1.114, 95% CI: 0.896–1.385, *p* = 0.332), CAM (HR 0.710, 95% CI: 0.562–0.896, *p* = 0.004), NIV (HR 0.704, 95% CI: 0.603–0.822, *p* < 0.001), and PEM (HR 0.868, 95% CI: 0.794–0.950, *p* = 0.002). Differences between individual agents were significant (*p* = 0.003). In Figure 2d, the treatment line of subgroup analysis was conducted. All treatment lines studied were either first (HR 0.820, 95% CI: 0.641–1.049, *p* = 0.115), second (HR 0.819, 95% CI: 0.688–0.974, *p* = 0.024), or third (HR 0.892, 95% CI: 0.701–1.134, *p* = 0.349). There was no significant difference between treatment lines (*p* = 0.839). In the subgroup of tumor site, there was no difference between groups (*p* = 0.131). In addition, esophageal cancer (HR 0.774, 95% CI: 0.671–0.892, *p* < 0.001) and G/GEJ cancer (HR 0.902, 95% CI: 0.784–1.038, *p* = 0.151) had lower HR compared to controls (Figure 2e). Overall, analyses indicated high heterogeneity.

#### 3.3.2. PFS

For secondary efficacy outcome, a total of seven studies with a population of 3700 were included [24,25,26,27,28,29,30] in the analysis of the HR of PFS. As shown in Figure 3a, the overall PFS of the patients in the intervention group had lower HR than that of the control group (HR 0.991, 95% CI: 0.778–1.263, I^2^ = 90.727%, *p* = 0.942). Regarding subgroup analysis for PD-1 and PD-L1 inhibitors, no significant results were observed between the groups (*p* = 0.174) of anti-PD-1 and anti-PD-L1 (HR 0.933, 95% CI: 0.727–1.197, *p* = 0.585 vs. HR 1.530, 95% CI: 1.246–2.986, *p* = 0.213, respectively, Figure 3b). Upon evaluating individual agents (Figure 3c), differences among them were observed as follows: AVE (HR 1.530, 95% CI: 0.741–3.159, *p* = 0.250), CAM (HR 0.690, 95% CI: 0.337–1.412, *p* = 0.310), NIV (HR 0.935, 95% CI: 0.614–1.425, *p* = 0.756), and PEM (HR 1.026, 95% CI: 0.683–1.541, *p* = 0.903). The difference between each agent was not significant (*p* = 0.483). In the subgroup of treatment line, no significantly differences were observed between the groups (*p* = 0.413) for first (HR 0.650, 95% CI: 0.334–1.265, *p* = 0.205), second (HR 1.054, 95% CI: 0.778–1.427, *p* = 0.734), and third (HR 1.057, 95% CI: 0.654–1.709, *p* = 0.819) (Figure 3d). As shown Figure 3e, the tumor site was not significantly different between groups (*p* = 0.231). The tumor site was esophageal cancer and G/GEJ cancer (HR 0.886, 95% CI: 0.659–1.192, *p* = 0.425 vs HR 1.190, 95% CI: 0.814–1.739, *p* = 0.990), respectively. 

#### 3.3.3. SAE

For safety, four studies were included [24,27,28,30], and a population size of 2109 was used in the analysis. As observed in the forest plot analysis, the intervention group had 1.033 times upper odds for SAE than the control (Figure 4). Moreover, the results showed that SAEs were not significantly different between the intervention and control groups. (OR 1.033, 95% CI: 0.717–1.488, I^2^ = 55.776%, *p* = 0.861).

### 3.4. Risk of Bias and Strength of Evidence

Deviations from intended intervention bias items were a major concern for all eight studies [24,25,26,27,28,29,30,31]. Regarding the bias in the outcome measurement, two studies were at high risk, while the remaining were of some concern. The details of the risk of bias for each study are shown in Figure 5. Table 2 illustrates the quality of evidence using the GRADEpro method for the effects of PD-1/PD-L1 inhibitors compared to the control in the outcome. Egger’s regression test suggested no evidence of publication bias (*p* = 0.819) (Figure 6).

## 4. Discussion

The present systematic review and meta-analysis was conducted to evaluate the efficacy and safety of PD-1 and PD-L1 inhibitors in patients with esophageal or G/GEJ cancer patients. Our findings suggested that both PD-1 and PD-L1 inhibitor use was associated with increased OS in these patients, as compared to the PFS. As surrogate endpoints for detecting the benefits of anticancer therapy, OS and PFS are considered to be of prime importance for drug approval. One of the first immunotherapies approved by the FDA in 2011, ipilimumab, showed extended survival among patients with advanced melanoma. To guarantee the approval of drugs and provide one of the most direct measures of true clinical benefit, prolonging the efficacy of survival rates prove most useful in identifying promising agents for cancer treatment [32]. Despite taking into consideration various confounders resulting from biases, survival is an important prognostic measure for more advanced-stage cancers [33]. However, although both the OS and PFS are considered meaningful efficacy endpoints of anticancer therapy, including traditional immunotherapy, the correlation between these two has not been sufficiently demonstrated. According to Blumenthal et al., there was no association between the OS and PFS (coefficient determination, R^2^ = 0.08) in the case of targeted and standard therapies in advanced non-small-cell lung cancer treatment [34]. Similarly, Mushti et al. also demonstrated a weak association between the PFS and OS in both trial and individual-level analyses in immunotherapy, particularly with regard to PD-1 and PD-L1 inhibitors (R^2^ = 0.1303 and R^2^ = 0.1277, respectively). Thus, it was concluded that PFS could not prove a sufficient surrogate endpoint to replace the OS in assessing the clinical benefit in immuno-oncology trials [35]. Furthermore, pathophysiological differences in the amplification of chromosome 11q13 and microsatellite instability in advanced esophageal cancer might contribute to beneficial effects of the OS rather than the PFS parameters, when treated with PD-1/PD-L1 inhibitors [9]. As the most direct measure of clinical benefit, outcomes of the OS post-treatment with PD-1/PD-L1 inhibitors in trials with esophageal or G/GEJ cancer patients were incongruous. Although previous meta-analyses consistently exhibited improvements in the OS rather than the PFS in esophageal or G/GEJ cancer treatment with PD-1/PD-L1 inhibitor, limitations such as the lack of controlled data or the possibility of known/unknown biases increasing heterogeneity in the analysis might result in less statistically powerful conclusions [6,9]. Furthermore, a previous study included only two studies in the analysis demonstrating improvement of the OS and PFS in the control group, rather than the anti-PD-1 or anti-PD-L1 groups, to treat esophageal or G/GEJ cancer patients [36]. One meta-analysis study that included data but only provided by abstract limited the preciseness of the outcomes [13]. Thus, in terms of clinically meaningful surrogate endpoints to estimate the activity of anti-PD-1 and anti-PD-L1 in treating advanced cancers such as esophageal or G/GEJ cancer, as the current study showed, improving the OS post-treatment with PD-1/PD-L1 inhibitors could provide important information on optimal treatment modalities.

In addition, the current study demonstrated that PD-1 inhibitor usage had a stronger association with improved OS in esophageal or G/GEJ cancer treatment than did PD-L1 inhibitors. In addition to blocking PD-1 and PD-L1 as well as reinvigorating T-cell activity, PD-1 inhibitors competitively bind to PD-1 along with PD-L1 by sharing overlapping binding surfaces [37]. Compared to PD-1 inhibitors, PD-L1 inhibitors are usually not associated with significant conformational changes in PD-L1 [13]. Moreover, such discrepancies might be related to the substantial variations in the mechanisms of action of a single PD-1/PD-L1 blockade agent [15,38]. Based on discrepancies in the mechanisms of action between PD-1/PD-L1 inhibitors, more attention is being paid to the differences between anti-PD-1 and anti-PD-L1 treatments in clinical practice, and evidence-based analysis to understand their comparable efficacies is urgently required [15]. Taking the above into consideration, an evaluation of the differences in response efficacy of anti-PD-1 and anti-PD-L1 usage in various cancer types was conducted in the present study. To this end, better objective response rates and durations of response were observed when treatment with PD-1 inhibitor was carried out [35]. Although Koneru et al. indirectly compared the safety and efficacy of PD-1 and PD-L1 antibodies across solid tumors and reported no significant differences [39], a prior meta-analysis with head-to-head comparison nonetheless demonstrated favorable OS outcomes of PD-1 inhibitors in cancer treatment [40]. Although the current outcome specifically supports PD-1 inhibitors as optimal treatment for esophageal or G/GEJ cancer rather than PD-L1, considering the limited number of studies conducted with PD-L1, more caution should be applied in clinical practice [41].

As to the safety issues associated with the administration of PD-1/PD-L1 inhibitors in esophageal or G/GEJ cancer patient treatment, the current study elucidated no significant differences of developing SAEs in the anti-PD-1/PD-L1 therapy group as compared to control groups. Enhancing the immune response against cancer cells, particularly in the case of adverse events—called immune-related adverse events (irAEs)—resulting from immune checkpoint inhibitors, such as PD-1/PD-L1 inhibitors, should be a primary focus [42]. Such toxicities are different from those caused by traditional chemotherapy, which affects whole cells in the body, resulting in significant reactions such as blood cell reduction or abnormal gastrointestinal, liver, and renal functions [43]. Although PD-1/PD-L1 inhibitors as immune checkpoint blockers can cause a widespread immune response affecting several organs, targeting only the immune system without affecting other cells in the body could possibly reduce the severity of these adverse events [44]. A prior meta-analysis involving 3450 patients also demonstrated a higher prevalence of irAEs (such as all-grade rash, pruritus, or hyperthyroidism) with a generally lower risk of adverse events and treatment discontinuation compared to chemotherapy [45]. When evaluating the risks and benefits of SAEs, PD-1/PD-L1 inhibitor use could be more beneficial in treating patients with esophageal or G/GEJ cancer.

The present study had several limitations. Firstly, to compare efficacy, the OS, and the PFS, more studies need to be performed with each subgroup. A lack of studies may cause uncertain biases while interpreting results from the analyses [11]. However, recently, more studies in various settings have been performed, which could not meet the inclusion criteria of the current study. Thus, we expect that more updated information of OS or PFS relevant to using anti-PD-1 and PD-L1 will be provided by pivotal studies soon [4,28,46]. Secondly, the control groups of the included studies received different types of chemotherapies. Since different SOCs were indicated in clinical practice guidelines such as National Comprehensive Cancer Network (NCCN) guidelines depending on the patients’ status, various chemotherapies were dosed to the control groups in the included studies.

Furthermore, this study did not assess the cost-effectiveness of PD-1/PD-L1 inhibitor usage in treating patients with esophageal or G/GEJ cancer. However, this topic is beyond the scope of this work and can be further investigated in subsequent studies. Finally, ethnic or regional variations were not evaluated in the current study. Although such differences play a role in the development of gastric cancer [1], in general, the efficacy and toxicities of PD-1/PD-L1 inhibitors do not vary drastically across geographic regions or races [41]. Hence, more studies evaluating these differences in the use of PD-1/PD-L1 inhibitors are necessary.

## 5. Conclusions

In conclusion, our analysis revealed that PD-1/PD-L1 inhibitors significantly prolonged the OS as compared to the control, while no significant effect on the PFS was observed in patients with esophageal or G/GEJ cancer. Compared to the PD-L1 inhibitors, PD-1 inhibitors were more strongly associated with improved OS in these patients. Coming to the development of SAEs, PD-1/PD-L1 inhibitor usage did not contribute to an increase in SAEs in esophageal or G/GEJ cancer. Although limited improvement of the PFS was observed, as a measure of efficacy, the OS increment associated with PD-1/PD-L1 inhibitor use might provide important information for determining the optimal treatment for esophageal or G/GEJ cancer. Furthermore, considering the significant intervention-specific differences shown in the subgroup analysis, more individualized therapies should be implemented in clinical practice.

## Figures and Tables

**Figure 1 jcm-10-03612-f001:**
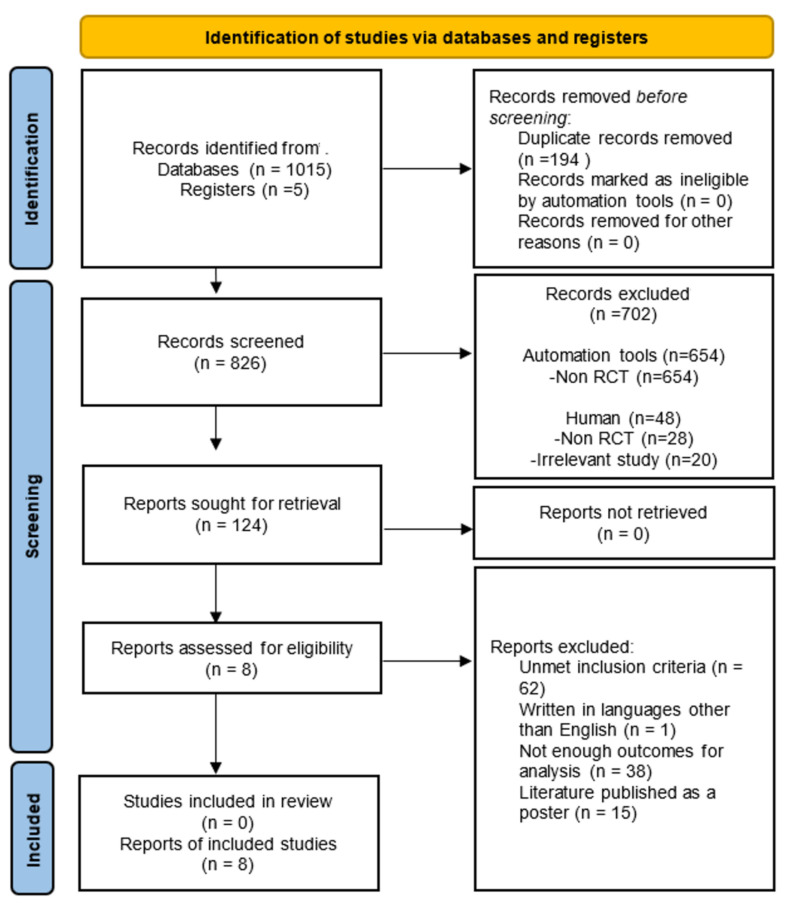
The PRISMA flowchart of the study selection process for the meta-analysis.

**Figure 2 jcm-10-03612-f002:**
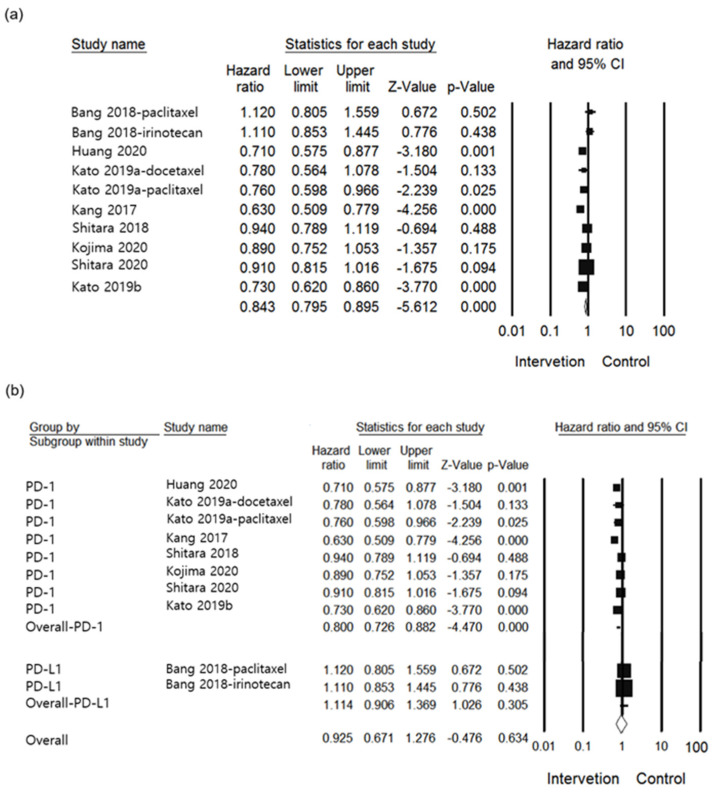
Forest plot of the efficacy of PD-1 /PD-L1 inhibitor use in esophageal or G/GEJ cancer patients: (**a**) Overall OS; (**b**) Subgroup analysis of compared between anti-PD-1 and PD-L1of OS; (**c**) Subgroup analysis of individual agents of OS; (**d**) Subgroup analysis of treatment line of OS; (**e**) Subgroup analysis of tumor site of OS (AVE: avelumab; CAM: camrelizumab; G/GEJ: gastric and gastroesophageal junction; NIV: Nivolumab; PEM: Pembrolizumab; PD-1: PD-1 inhibitor; PD-L1: PD-L1 inhibitor).

**Figure 3 jcm-10-03612-f003:**
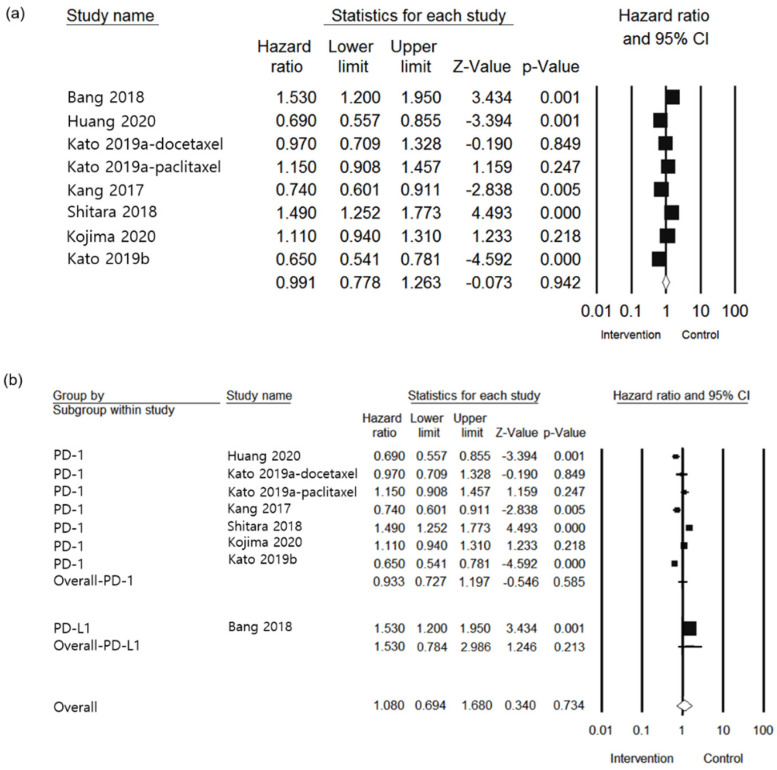
Forest plot of the efficacy of PD-1/PD-L1 inhibitor use in esophageal or G/GEJ cancer patients: (**a**) Overall PFS; (**b**) Subgroup analysis comparison between anti-PD-1 and PD-L1 of PFS; (**c**) Subgroup analysis of individual agents of PFS; (**d**) Subgroup analysis of treatment line of PFS; (**e**) Subgroup analysis of tumor site of PFS (AVE: avelumab; CAM: camrelizumab; G/GEJ: gastric and gastroesophageal junction; NIV: nivolumab; PEM: pembrolizumab; PD-1: PD-1 inhibitor; PD-L1: PD-L1 inhibitor).

**Figure 4 jcm-10-03612-f004:**
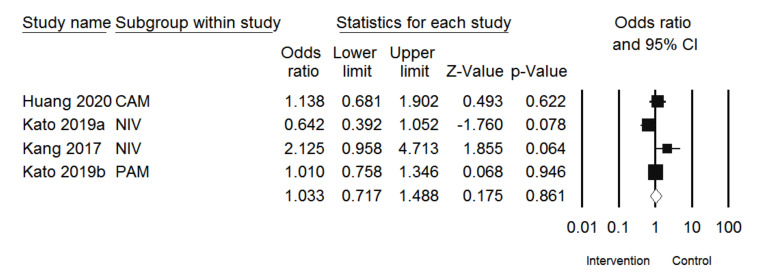
Forest plot of SAE developments after treating PD-1/PD-L1 inhibitors in esophageal or G/GEJ cancer patients (CAM: camrelizumab; NIV: nivolumab; PEM: pembrolizumab).

**Figure 5 jcm-10-03612-f005:**
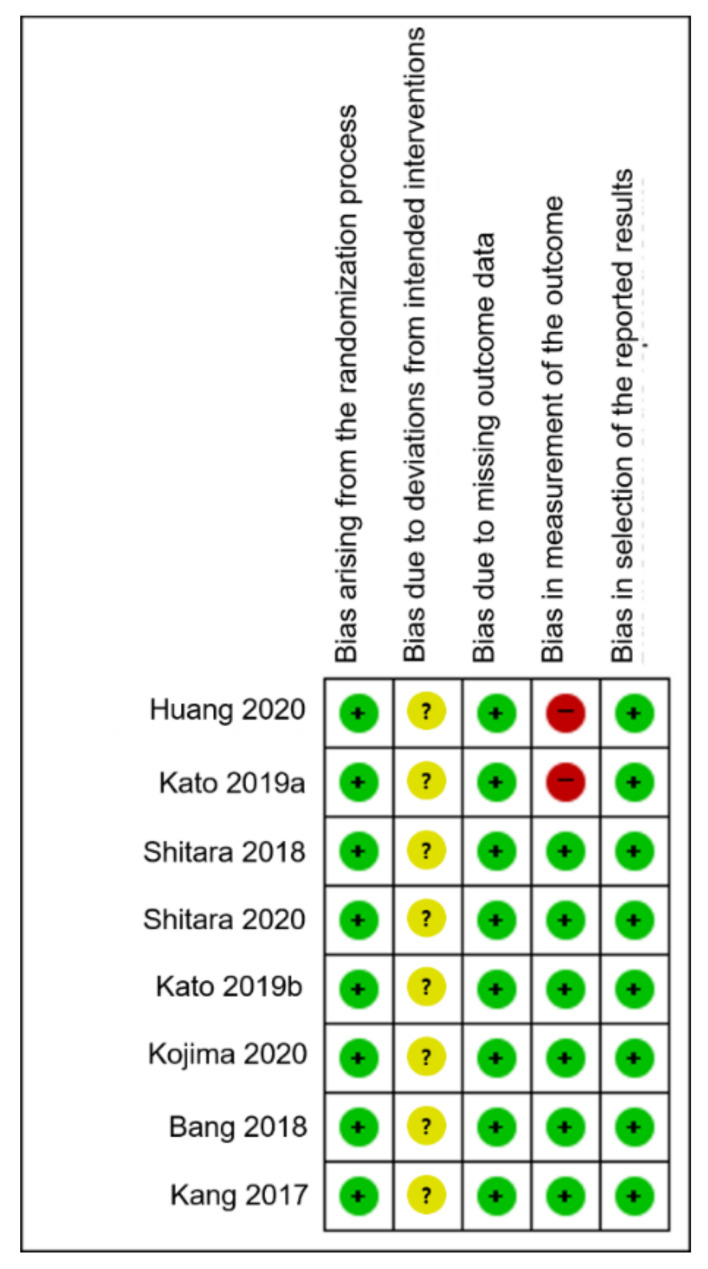
Risk of bias assessment. The colors of symbols were explained with that risk of bias. High, low, and unclear risks were marked as red, green, and yellow.

**Figure 6 jcm-10-03612-f006:**
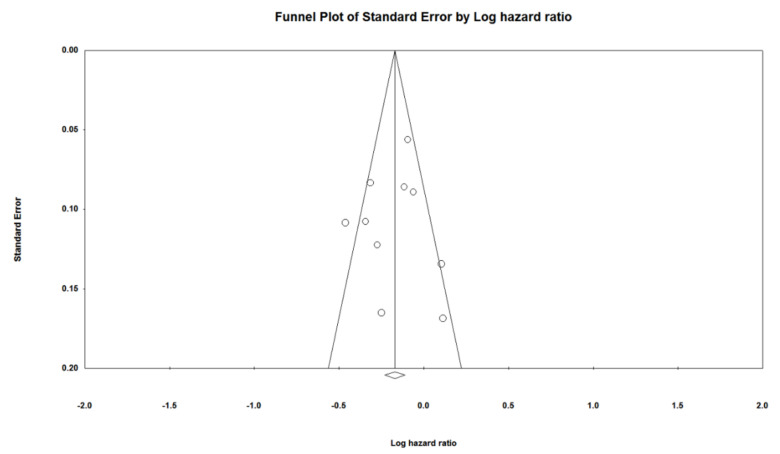
Funnel plot for assessing publication of bias. Open circles suggested included studies and diamond explained observed effect size.

**Table 1 jcm-10-03612-t001:** Characteristics of included studies.

Author, Publication Year, Ref	Trial	Tumor	ClinicalTrials.gov Number	PD-1/PD-L1 Inhibitor	Control	Study Design	Phase	Patient Number (*n*)		Treatment Line	Primary Endpoint of Included Study	Intervention Methods	Used Outcomes
								Intervention	Control				
Kang 2017 [24]	ATTRACTION-2	G/GEJ	NCT02267343	Nivolumab	Placebo	RCT	III	330	163	Third line	OS	Nivolumab 3 mg/kg Q2W ^a^	OS, PFS, SAE
Shitara 2018 [25]	KEYNOTE-061	G/GEJ	NCT02370498	Pembrolizumab	Paclitaxel	RCT	III	296	296	Second line	OS	Pembrolizumab 200 mg Q3W ^b^	OS, PFS
Bang 2018 [26]	JAVELIN Gastric 300	G/GEJ	NCT02625623	Avelumab	Paclitaxel or Irinotecan	RCT	III	185	186	Third line	OS	Avelumab 10 mg/kg Q2W ^a^	OS, PFS
Kato 2019a [27]	ATTRACTION-3	Esophageal	NCT02569242	Nivolumab	Paclitaxel + Docetaxel	RCT	III	210	209	Second line	OS	Nivolumab 240 mg Q2W ^a^	OS, PFS, SAE
Kato 2019b [28]	KEYNOTE-590	Esophageal	NCT03189719	Pembrolizumab	Placebo	RCT	III	373	376	First line	OS	Pembrolizumab 200 mg Q3W ^b^	OS, PFS, SAE
Kojima 2020 [29]	KEYNOTE-181	Esophageal	NCT02564263	Pembrolizumab	Paclitaxel, Docetaxel or Irinotecan	RCT	III	314	314	Second line	OS	Pembrolizumab 200 mg Q3W ^b^	OS, PFS
Huang 2020 [30]	ESCORT	Esophageal	NCT03099382	Camrelizumab	Docetaxel or Irinotecan	RCT	III	228	220	Second line	OS	Camrelizumab 200 mg Q2W ^a^	OS, PFS, SAE
Shitara 2020 [31]	KEYNOTE-062	G/GEJ	NCT02494583	Pembrolizumab	Cisplatin + Fluorouracil or Capecitabine	RCT	III	256	250	First line	OS	Pembrolizumab 200 mg Q3W ^b^	OS

G/GEJ: Gastric or gastro-oesophageal junction NA: Not available; OS: Overall survival; PFS: Progression-free survival; RCT: Randomized controlled trials; SAE: Serious adverse event; ^a^ Once every 2 weeks; ^b^ Once every 3 weeks.

**Table 2 jcm-10-03612-t002:** Summary of findings for outcome comparing PD-1/PD-L1 inhibitors to control group based on the GRADE approach.

Outcomes	No. of Participants (Studies)	Risk of Bias	Inconsistency	Indirectness	Imprecision	Other Considerations	Effect (95% CI)	Quality of Evidence
OS								
Overall	3738 (8)	Not serious	Not serious	Not serious	Not serious	None	HR 0.843 (0.795, 0.895)	⨁⨁⨁⨁ HIGH
Anti-PD-1	3835 (7)	Not serious	Not serious	Not serious	Not serious	None	HR 0.822 (0.773, 0.875)	⨁⨁⨁⨁ HIGH
Anti-PD-L1	371 (1)	Not serious	Not serious	Not serious	Not serious	None	HR 1.114 (0.906, 1.369)	⨁⨁⨁⨁ HIGH
Avelumab	371 (1)	Not serious	Not serious	Not serious	Not serious	None	HR 1.114 (0.906, 1.369)	⨁⨁⨁〇 MODERATE
Camrelizumab	448 (1)	Not serious	Not serious	Not serious	Not serious	None	HR 0.710 (0.575, 0.877)	⨁⨁⨁⨁ HIGH
Nivolumab	912 (4)	Not serious	Not serious	Not serious	Not serious	None	HR 0.702 (0.608, 0.810)	⨁⨁⨁⨁ HIGH
Pembrolizumab	1847 (4)	Not serious	Not serious	Not serious	Not serious	None	HR 0.872 (0.811, 0.938)	⨁⨁⨁⨁ HIGH
Treatment line (first)	1255 (2)	Not serious	serious	Not serious	Not serious	None	HR 0.849 (0.775, 0.931)	⨁⨁⨁〇 MODERATE
Treatment line (second)	2087 (3)	Not serious	Not serious	Not serious	Not serious	None	HR 0.836 (0.763, 0.917)	⨁⨁⨁⨁ HIGH
Treatment line (third)	864 (2)	Not serious	serious	Not serious	Not serious	None	HR 0.845 (0.729, 0.980)	⨁⨁⨁〇 MODERATE
Tumor site (esophageal)	2244 (4)	Not serious	Not serious	Not serious	Not serious	None	HR 0.778 (0.710, 0.851)	⨁⨁⨁⨁ HIGH
Tumor site (G/GEJ)	1962 (4)	Not serious	serious	Not serious	Not serious	None	HR 0.897 (0.829, 0.971)	⨁⨁⨁〇 MODERATE
PFS								
Overall	3700 (7)	Not serious	Very serious	Not serious	Not serious	None	HR 0.993 (0.923, 1.067)	⨁⨁〇〇 LOW
Anti-PD-1	3329 (6)	Not serious	serious	Not serious	Not serious	None	HR 0.951 (0.881, 1.026)	⨁⨁⨁〇 MODERATE
Anti-PD-L1	371 (1)	Not serious	Not serious	Not serious	Serious	None	HR 1.530 (1.200, 1.950)	⨁⨁⨁〇 MODERATE
Avelumab	371 (1)	Not serious	Not serious	Not serious	Serious	None	HR 1.530 (1.200, 1.950)	⨁⨁⨁〇 MODERATE
Camrelizumab	448 (1)	Not serious	Not serious	Not serious	Not serious	None	HR 0.690 (0.557, 0.855)	⨁⨁⨁⨁ HIGH
Nivolumab	912 (2)	Not serious	Serious	Not serious	Not serious	None	HR 0.911 (0.792, 1.048)	⨁⨁⨁〇 MODERATE
Pembrolizumab	1969 (3)	Not serious	Not serious	Not serious	Not serious	None	HR 1.044 (0.944, 1.154)	⨁⨁⨁⨁ HIGH
Treatment line (first)	749 (1)	Not serious	Not serious	Not serious	Serious	None	HR 0.650 (0.541, 0.781)	⨁⨁⨁〇 MODERATE
Treatment line (second)	2161 (4)	Not serious	Serious	Not serious	Not serious	None	HR 1.097 (1.001, 1.203)	⨁⨁⨁〇 MODERATE
Treatment line (third)	864 (2)	Not serious	Serious	Not serious	Not serious	None	HR 1.006 (0.859, 1.179)	⨁⨁⨁〇 MODERATE
Tumor site (esophageal)	2244 (4)	Not serious	Serious	Not serious	Not serious	None	HR 0.880 (0.802, 0.965)	⨁⨁⨁〇 MODERATE
Tumor site (G/GEJ)	1456 (3)	Not serious	Serious	Not serious	Serious	None	HR 1.202 (1.069, 1.351)	⨁⨁〇〇 LOW
SAE	2147 (5)	Not serious	Serious	Not serious	Not serious	None	OR 0.983 (0.795, 1.217)	⨁⨁⨁〇 MODERATE

CI: confidence interval; G/GEJ: gastric and gastroesophageal junction; HR: hazard ratio; OR: odds ratio; PD-1: programmed death 1; PD-L1: PD ligand 1; ⨁ = attainment of Grading of Recommendations, Assessment, Development, and Evaluation criteria.; 〇 = uncertainty of attaining Grading of Recommendations, Assessment, Development, and Evaluation criteria.

## Data Availability

The data used for the findings of the current study are available on request from the corresponding author.

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
