# Peer review of "Comparative Impact of PD-1 and PD-L1 Inhibitors on Advanced Esophageal or Gastric/Gastroesophageal Junction Cancer Treatment: A Systematic Review and Meta-Analysis"

_jcm, 2021, doi:10.3390/jcm10163612_

Round 1
Reviewer 1 Report
This meta-analysis by Oh and colleagues aims at evaluating the effectiveness of anti-PD-1 and anti-PD-L1 inhibitors in gastroesophageal cancer.
There are few major comments.
- This review focuses on gastroesophageal cancer, however it is not clear the exact tumor site object of this work (gastric and gastroesophageal junction cancer? only gastroesophageal junction cancer? also esophageal - even squamous - cancer?). Theoretically, different tumors (such as esophageal and gastric cancer) might not be together considered. This aspect should be better clarified and justified.
- It is not mentioned the setting where the immune checkpoint inhibitors are used (line of treatment). I hypothesize that different studies were together considered given the relative limited number of studies. However, it must be discussed.
- Figures 2 and 3: the legends and the figures can not be easily interpreted.
- A comparison between anti-PD-1 and PD-L1 can not be statistically performed in this way.
- The control group of different studies may vary. Discuss this aspect.
- The authors considered OS and PFS of various studies, considering them as major outcomes. However, they did not represent primary endpoints of the singles studies and this could represent a bias.
Author Response
We uploaded a file for responses to the reviewer's comments.

Reviewer 2 Report
the manuscript is well written though there is missing data and i would like that the authors will add the data from keynote 570 chekmate 577 and chekmate 649 for the complete this meta analysis this studies are the pivotal studies that need to be included in this kind of manuscript
Author Response
We uploaded a file of responses to the reviewer's comments.

Round 2
Reviewer 1 Report
I appreciated to review a revised version of the the manuscript titled "Comparative impact of PD-1 and PD-L1 inhibitors on advanced gastric esophageal cancer treatment: A systematic review and meta-analysis" submitted by SuA Oh and colleagues for consideration in JCM.
I would like to thank the authors for taking the time to thoroughly respond to my comments and carefully address them in the revised version of the manuscript. I also would like to congratulate them for this major revision of their work. Overall, the major limitations that I had highlighted are now acknowledged and addressed.
Reviewer 2 Report
i thank the authors for adding the data that was missing to analysis and to references list
i would change the title of the manuscript and omit he world meta analysis systematic review is the appropriate
